# Pertussis Outbreak During 2023 in Gipuzkoa, North Spain

**DOI:** 10.3390/vaccines12101192

**Published:** 2024-10-18

**Authors:** José María Marimón, Milagrosa Montes, Nahikari Vizuete, Lorea Alvarez Guerrico, Adrian Hugo Aginagalde, Alba Mir-Cros, Juan José González-López, Diego Vicente

**Affiliations:** 1Biogipuzkoa Health Research Institute, Infectious Diseases Area, Microbiology Department, Osakidetza Basque Health Service, Donostialdea Integrated Health Organization, Donostia University Hospital, 20014 Donostia-San Sebastian, Spain; mariamilagrosa.montesros@osakidetza.eus (M.M.); diego.vicenteanza@osakidetza.eus (D.V.); 2Sub-Directorate for Public Health and Addictions of Gipuzkoa, Ministry of Health of the Basque Government, 20013 Donostia-San Sebastian, Spainl-alvarezguerrico@euskadi.eus (L.A.G.); ah-llorente@euskadi.eus (A.H.A.); 3Epidemiology of Chronic and Communicable Diseases Group, Biogipuzkoa Health Research Institute, 20014 Donostia-San Sebastian, Spain; 4Department of Clinical Microbiology, Vall d’Hebron Hospital Universitari, Passeig Vall d’Hebron 119-129, 08035 Barcelona, Spain; alba.mir@vhir.org (A.M.-C.); juanjo.gonzalez@vallhebron.cat (J.J.G.-L.); 5CIBER de Enfermedades Infecciosas (CIBERINFEC), Instituto de Salud Carlos III, 28029 Madrid, Spain; 6Department of Genetics and Microbiology, Universitat Autònoma de Barcelona, 08193 Bellaterra, Spain; 7Faculty of Medicine, University of the Basque Country, UPV/EHU, 20014 Donostia-San Sebastian, Spain

**Keywords:** *Bordetella pertussis*, vaccination, outbreak

## Abstract

Background: Pertussis has re-emerged in many countries despite the wide use of vaccines for over 60 years. During 2023, we observed an increase in the incidence of pertussis in Gipuzkoa, north of Spain (with a population of 657,140 inhabitants), mainly affecting children between 11 and 15 years of age. Methods: This study included all confirmed cases diagnosed by PCR in nasopharyngeal swab samples. The genome of seven isolates collected in 2023 was sequenced. Results: Between 2018 and 2023, 884 cases of whooping cough were diagnosed. Pertussis incidence (in cases per 100,000 inhabitants) decreased from 36.7 in 2018 to no cases in 2021, increasing again to 56.8 in 2023. In 2023, the age group of 11–15 years old had the highest incidence rate of 409.3. Only 2 of the 56 children < 6 years old required hospitalization, and there were no deaths. The seven isolates collected in 2023 showed the same BPagST-4 (*ptxA1/ptxP3/prn2/fim2-1/fim3*-1 allelic combination), with all of them expressing the pertactin antigen. Conclusions: Immunity waning after the last dose of vaccination at 6 years old, together with the lack of circulation of *Bordetella pertussis* during the COVID-19 pandemic, were probably the main reasons for the high increase in the incidence of pertussis in Gipuzkoa in 2023.

## 1. Introduction

Pertussis, or whooping cough, is an infection of the upper respiratory tract characterized by a violent cough that can last for weeks, also known as “the cough of the 100 days”. It is primarily caused by *Bordetella pertussis*, although other *Bordetella* species, such as *Bordetella parapertussis* and *Bordetella holmesii*, can also be responsible [1,2]. *B. pertussis* is exclusively transmitted from human to human by aerosol droplets. Pertussis can affect people of all ages, but children under 6 months of age are at high risk of serious complications and hospitalizations [3].

Pertussis was first described in the epidemic of Paris in 1578 [4]. Until the first vaccines appeared at the beginning of the 1940s, pertussis was a frequent cause of infant morbidity and mortality, with a fatality rate of around 10% [3]. The first vaccines were composed of the whole bacterial cell as the antigen and showed an effectiveness of 70% to 90% in the prevention of severe pertussis. However, it provoked pain and swelling at the site of the injection as well as systemic events such as fever in up to 50% of doses [5]. In Spain, pertussis vaccination with the whole-cell vaccine (WCV) was introduced in the 1960s but since 1998 was progressively replaced by the acellular pertussis vaccine (ACV). The current ACV contains two to five antigens; all of them include the pertussis toxin (PT) and the filamentous hemagglutinin (FHA); the other antigens pertactin (PRN) and fimbria type 2 and 3 (FIM2/3) are present depending on the manufacturer.

In Gipuzkoa, Basque Country, north of Spain, all pertussis vaccines used since 2004 have been ACVs [6]. Currently, two kinds of ACVs are used, always in combination with diphtheria (D) and tetanus (T) vaccines according to the antigen load: high antigen load (DTaP) vaccines, only used in primary vaccination, and reduced diphtheria and pertussis antigens (dTap), used in the booster doses. The current pertussis vaccination schedule in the Basque Country consists of two primary doses of a hexavalent vaccine that combines DTaP, inactivated poliovirus (IPV), *Haemophilus influenzae* b (Hib) conjugate, and hepatitis B (HepB) vaccines administered at 2 and 4 months of age, and two booster doses at 11 months with the hexavalent vaccine and at 6 years with a tetravalent vaccine, which combines DTaP and IPV vaccines. Since 2015, a recommendation for dTpa vaccination in each pregnancy has been indicated between 27 and 31 weeks to protect the newborn with maternal antibodies. Since 2017, primary childhood vaccination in the Basque Country has a coverage rate of over 90%, while in pregnant women, coverage rates vary between 62% and 89% [7].

Despite the high vaccination coverage rate in many countries, there are cases of whooping cough reported from time to time, with outbreaks or peak of infections occurring every 3–5 years. In Spain, there have been five epidemic waves of pertussis between 1998 and 2018 [8]. In this work, we describe the increase in the number of pertussis cases occurring in Gipuzkoa, northern Spain, during the spring of 2023 and the characterization by whole-genome sequencing (WGS), the identification of vaccine antigens, and the molecular determination of susceptibility to macrolides of the isolates causing the outbreak.

## 2. Material and Methods

### 2.1. Case Definition

Only confirmed cases, as defined by the National Epidemiological Surveillance Network (RENAVE) case definition [9] were included: people who meet both the clinical (cough lasting for at least two weeks and one or more signs of these three: paroxysmal cough, inspiratory stridor, or vomiting caused by cough) and laboratory criteria (culture or PCR positive to *B. pertussis* or specific antibody response to *B. pertussis*).

Population data were obtained from the official data of the Basque Statistics Institute-EUSTAT (www.eustat.eu).

### 2.2. Sampling

From 2018 to 2023 in Gipuzkoa, north of Spain, a nasopharyngeal swab using a Dacron flocked swab (Copan Diagnostics, Murrieta, CA, USA) was collected from all patients with a clinical suspicion of *B. pertussis* infection and sent to the Microbiology Department of Hospital Universitario Donostia (HUD) for testing. All swabs were sent at room temperature using the UTM-RT medium (COPAN, Brescia, Italy). As these transport media contain antibiotics, a second nasopharyngeal swab was collected using a cotton swab (DELTALAB, Barcelona, Spain) and sent in an AMIES transport medium to avoid the inhibition of bacterial growth.

### 2.3. Vaccination Status

The vaccination status of the patients with pertussis diagnosed in 2023 was obtained from the Vaccination Information System (SIV) of the Health Department of the Basque Country.

### 2.4. Microbiological Test

*B. pertussis* was detected in the samples of all patients using a commercial real-time PCR (Viasure, Certest BIOTEC, SL, Zaragoza, Spain) that detects and differentiates *B. pertussis*, *B. parapertussis*, and *B. holmesii* by targeting the insertion sequence (IS) 481 for *B. pertussis* and *B. holmesii*, hIS1001 for *B. holmesii* and a region of the pIS1001 for *B. parapertussis*. *B. pertussis* was cultured using *Bordetella* charcoal agar plates (MAIM, Barcelona, Spain) that were incubated in a moist chamber at 35 °C for 14 days at room atmosphere. Characteristic colonies (small, shiny, greyish round colonies) that grew after a minimum of three days of incubation were identified by matrix-assisted laser desorption ionization–time of flight mass spectrometry (MALDI-TOF MS, Bruker Daltonics, Germany). MALDI-TOF was performed using direct plate testing. After drying the colony, 2 µL of matrix dissolved in acetonitrile was added and allowed to dry again. Analysis was performed using the MALDI biotyper sirius system (Bruker Daltonik GmbH, Bremen, Germany) and the MBT Compass MSP library v. 4.1. For species level identification, a cut-off score of ≥2.0 was used.

The genomic mutation A2047G in domain V of the 23S rRNA gene conferring macrolide resistance was studied in all *B. pertussis* PCR-positive respiratory samples by PCR sequencing as described [10]. In addition, an erythromycin gradient strip test (0.016 µg/mL to 256 µg/mL) (E-test, bioMerieux, Marcy l’Étoile, France) was performed on the 19 isolates that were cultured.

### 2.5. Antigen Production

The production of PT, FHA, PRN, and the fimbrial serotype was carried out using an indirect whole-cell ELISA with specific antibodies (99/512 for PT S1 subunit, 99/572 for FHA, 97/558 for PRN, 06/124 for FIM2, and 06/128 for FIM3; National Institute for Biological Standards and Control, https://www.nibsc.org) as previously described [1,11].

### 2.6. Whole Genome Sequencing

A single colony was subcultured on the same *Bordetella* charcoal agar plates, and, after growth for 24–48 h in a moist chamber at 35 °C, the nucleic acids were extracted using Qiagen columns. The DNA was quantified using Qubit (Thermo Fisher scientific, Waltham, MA, USA) and adjusted to 6 ng/mL or 20 ng/mL for preparing libraries using Illumina (Illumina DNA Prep) or Oxford Nanopore (ONT) kits (rapid barcoding kit). Sequencing was performed on an iSeq and a MinionMk1C, respectively. Hybrid assemblies using Illumina and ONT reads were carried out using Unicycler (Version 0.5.0) and annotated using prokka (Version 1.14.6). A comparison of the genomes was carried out using the Snippy tool (Version 4.6.0)

A maximum likelihood tree based on SNP differences was constructed with IQ-Tree using the sequences of the eight isolates from Gipuzkoa and a selection of half of the isolates collected between 1986 and 2018 from a previous Spanish study [11] harbored in the ENA database (study accession PRJNA667582). After arranging the isolates by collection date, the sequences of every second isolate were selected to construct the tree. All of the software tools are available on the European Galaxy server (https://usegalaxy.eu/, accesed on 7 July 2024). The final tree was annotated using iTOL software (https://itol.embl.de/, accesed on 7 July 2024).

Alleles of the different analyzed genes were obtained using the *Bordetella* database and software available on the Pasteur webpage (https://bigsdb.pasteur.fr/bordetella/, accesed on 7 July 2024).

The *B. pertussis* vaccine antigens sequence types (BPagST), which groups the nine loci *ptxP*, *ptxABCDE*, *fhaB2400_5550*, *fim2*, and *fim3* was used to classify the isolates sequenced.

Assembled genomic sequences were deposited in GenBank under the accession numbers SAMN40551699 to SAMN40551706.

### 2.7. Ethics

The study was approved by the local ethic committee under study number DVA-PER-2023-01.

## 3. Results

### 3.1. Epidemiology

From 2018 to 2023, 884 cases of whooping cough were observed in Gipuzkoa, north of Spain (with a population of 657,140 inhabitants). After no cases in 2021 and only two cases observed in March and December 2022, whooping cough reappeared in 2023 (Table 1).

The first in of 2023 appeared sporadically in different areas of the province of Gipuzkoa during March (one case) and the beginning of April (three cases). But in the second half of April, a cluster of pertussis was detected in a school (69 cases up to mid-June), affecting children aged 8–16 years. From then on, the infection expanded to the entire region uninterruptedly during the whole year. During 2023, in Gipuzkoa, the incidence of pertussis reached 55.5 cases per 100,000 inhabitants, but in the surrounding provinces of Araba and Bizkaia, the re-emergence of pertussis was not as pronounced, with 13 cases in Araba (incidence 3.3 cases per 100,000 inhabitants) and 113 in Bizkaia (incidence 9.9 cases/100,000 population). The incidence of whooping cough in Gipuzkoa in 2023 was the highest in recent years, peaking at 92 cases in June and exceeding the rates recorded in the years prior to the COVID-19 pandemic.

During 2023, children between 11 and 15 years old (incidence rate 409.3 cases per 100,000 inhabitants) were the age group with the highest incidence of pertussis, followed by children 6–10 years old (267.8 cases per 100,000 inhabitants), children 1–5 years (208.3 cases per 100,000 inhabitants), and children < 1 year of age (97 cases per 100,000 inhabitants). In the age group of 11–15 years old, the incidence increased more than four times compared to pre-pandemic years. This situation was different from previous years, in which the highest incidence was recorded among infants and decreased with increasing age (Figure 1). In fact, a decrease from 388 to 97 cases per 100,000 inhabitants was observed in the incidence of pertussis in children < 1 year old between 2019 and 2023. There were no cases in children younger than 2 months.

In 2023, of the 56 children under 6 years old with whooping cough, only 2, aged 3 and 7 months, required hospitalization. There were no deaths from whooping cough.

### 3.2. Erythromycin Resistance

A bacterial culture was performed in 249 of the 371 pertussis episodes of 2023, and *B. pertussis* was isolated in 19 of them (7.6%). The A2047G mutation in domain V of the 23S rRNA gene associated with erythromycin resistance was not detected in any *B. pertussis* PCR-positive samples. The 19 cultured isolates showed an erythromycin MIC < 0.016 µg/mL.

### 3.3. Vaccination Status of Cases

The majority of the patients were vaccinated with a trivalent DTaP vaccine containing three *B. pertussis* antigens: PT, FHA, and PRN (Infanrix^®^, GSK, Madrid, Spain). In 2021, the vaccine was changed to a hexavalent vaccine (DTaP-IPV-Hib-HepB) (Vaxelis^®^, MCM Vaccine B.V., Leiden, The Netherlands) that contains 5 *B. pertussis* antigens PT, FHA, PRN, FIM2, and FIM3.

Out of the 371 patients diagnosed with pertussis in 2023, the vaccination status was investigated in 327 (88.1%). Of them, 31 were older than 50 years and were not vaccinated; of the 37 cases diagnosed in people 25–50 years old, 27 did not have their vaccine status registered; and 10 were not vaccinated. Of the 303 patients aged <25 years old, vaccine status was recorded for 286, of which 273 (95.5%) were correctly vaccinated according to their age (Table 2).

### 3.4. Vaccine Antigen Expression and WGS Analysis

The genome of seven *B. pertussis* isolates collected between March and July 2023 (isolates HUD-B1 to HUD-B7), along with two control isolates, were sequenced. One clinical isolate was previously collected in our hospital in July 1999 (isolate HUD-B8), and the other was *B. pertussis* ATCC9340.

The seven isolates of 2023 (HUD-B1 to HUD-B7) showed the same BPagST-4 (*ptxP3*, *ptxA1*, *ptxB1*, *ptxC4*, *ptxD1*, *ptxE4*, *fhaB1*, *fim2-1*, and *fim3-1*). They only showed between 1 and 6 SNPs of difference between them but 36 SNP, 2 deletions, and 1 insertion with HUD-B8, isolated in 1999. Isolates ATCC9340 and HUD-B8 showed a different allelic combination and were BPagST-75 and BPagST-9 (*ptxP3*, *ptxA1*, *ptxB1*, *ptxC4*, *ptxD1*, *ptxE4*, *fhaB1*, *fim2-1*, *fim3-2*), respectively.

In the phylogenetic tree, isolates HUD-B1 to HUD-B7 were grouped together (Figure 2), forming a differentiated cluster within isolates of Clade III (*fim3-1*) described in Spanish isolates from 2005 to 2018 [11]. On the other hand, the isolate HUD-B8, which was isolated in Gipuzkoa in 1999, was grouped with isolates of Clade II (*fim3-2*).

ELISA showed that the isolate of 1999 (HUD-B8) and the seven isolates of 2023 expressed PRN, FHA, and PT antigens. However, the seven isolates of 2023 expressed FIM2, while the isolate from 1999 expressed FIM3.

## 4. Discussion

Despite the wide use of vaccines spanning more than 60 years, pertussis has re-emerged in most countries, including Spain, especially in adolescents and infants linked to the introduction of ACVs [8,12,13,14]. Different events can explain this re-emergence of pertussis, including population immunity waning due to less circulation of the pathogen and vaccine scape due to the adaptation of circulating clones under ACV-mediated immune selection pressure [8,9].

In Europe, after a few years of limited circulation of *B. pertussis* during the COVID-19 pandemic, an increase in cases during 2023 and 2024 was observed [15]. In Spain, a similar situation was observed, with an incidence of pertussis of 5.6 cases per 100,000 inhabitants during 2023 and a higher increase at the beginning of 2024 to 97.5, with children 10–14 years old being the group with the highest incidence [16]. In Denmark, a similar increase in pertussis between August 2023 to February 2024 and especially in adolescents, was recently reported [17]. In Gipuzkoa, the vaccination coverage rate in the childhood vaccination schedule is >95%, with vaccines that have shown an efficacy of 86–88% [18,19], but in Denmark, there is no booster vaccination for adolescents. As immunity developed after vaccination or natural infection wanes over time, the high incidence rates in children aged 11–15 years reflect a reduction in the immunity in that age group, while younger children were still protected by the last vaccine dose at 6 years old or with antibodies transmitted by vaccinated pregnant women to newborns. The lack of circulation of *B. pertussis* in the first two years of the COVID-19 pandemic could have created a greater proportion of the susceptible population (“immunity debt”) [20] that favored the circulation of *B. pertussis* when the non-pharmaceutical interventions against SARS-CoV-2 transmission were ended.

Another explanation for the re-emergence of pertussis is the selection of genotypes that have fitness advantages under the pressure of mass vaccination [21]. In this way, the use of ACVs has been associated in the last few years with the spread of pertactin-deficient pertussis strains. In a study performed in nine different European countries in four periods, from 1998 to 2015, it was observed that the longer the period since the introduction of ACVs containing pertactin as one of the antigens, the higher the frequency of circulating pertactin-deficient isolates [22]. In Spain, pertactin-deficient *B. pertussis* strains emerged at the same time that ACVs was introduced and increased their prevalence since their appearance, suggesting that the use of the ACV probably drove the antigenic swift towards altered pertactin expression [11]. After ACV introduction, 38% of *B. pertussis* isolates in Spain were pertactin-deficient, most of them showing the *prn*::del (−292, 1340) mutation and pertactin inactivation by *IS481* insertion [11]. Contrarily, the seven isolates analyzed from the outbreak in 2023 in Gipuzkoa formed a cluster that expressed the pertactin allele 2 (PRN2). The loss of natural immunity against PRN may have conditioned the current predominance of PRN-producing isolates. The PRN allele 2 was first described in isolates from the 1980s in Dutch isolates [23] that appeared driven by an immune selection of the WCV that contained the PRN allele 1 antigen. It was also found in 72% of isolates from the 1990s in Finland [24].

The profile of our isolates (BPagST-4) was the same and grouped together with isolates of Clade III, described in Spanish isolates during 2015–2018, exclusive of the ACV period [11] and is also the same as the most abundant profile A (52.3%) found in Austrian isolates between 2018 and 2020 [17] and in Norwegian isolates after ACV introduction [25]. It has also been observed in isolates from countries as far as South Africa [26], Iran [27] or China [28]. In South Africa, contrarily to Gipuzkoa, vaccine formulation did not have the pertactin antigen and in Iran, only WCV has been used since its introduction in the 1970s. The selection of the same clone in regions or countries using ACVs containing different antigens or even using WCV suggests that the vaccine composition does not have a determining effect on the selection of clones responsible for pertussis epidemics. It is possible that other bacterial virulence determinants, such as the *ptxP3* allele, which is thought to promote increased toxin production compared to other *ptxP* alleles, may enhance the transmission of *ptxP3* clones [29].

In Spain, pertussis is a notifiable disease. At the beginning of June 2023, after the outbreak in the school in May, a letter was sent to all pediatricians and general practitioners asking to take respiratory samples from patients with a clinical picture of pertussis. This alert, created in 2023 Gipuzkoa, probably helped detect most of the cases, as advisory alerts have been demonstrated to elevate reported pertussis cases and incidence, which may be underestimated due to a lack of suspicion of the disease or inadequate systematic testing [30]. This alert could also explain the difference in the lower incidences reported in nearby provinces that shared the same vaccination schedules.

This study has some limitations, the greatest of which is the low number of isolates genotyped, which reflects the difficulties in culturing this bacteria with particular nutritional requirements. However, the high genomic similarity of the isolates from 2023 and the great differences with the isolate of 1999 strongly suggest the presence of a predominant clone responsible for the outbreak in 2023.

## 5. Conclusions

It seems likely that the absence of *B. pertussis* circulation due to the lack of infections during the early years of the COVID-19 pandemic reduced the level of population immunity in Gipuzkoa [31]. This, together with the vaccine immunity waning during adolescence, made teenagers a population more susceptible to infection at the moment when a highly contagious clone is spread. The age of the majority of pertussis cases (5–15 years old), a consequence of the high vaccine coverage rate achieved in pregnant women and young children, was probably the main reason for the low number of admissions to hospitals with good outcomes. The ongoing adaptation of *B. pertussis* to vaccines makes necessary a continuous genomic surveillance of the strains causing disease.

## Figures and Tables

**Figure 1 vaccines-12-01192-f001:**
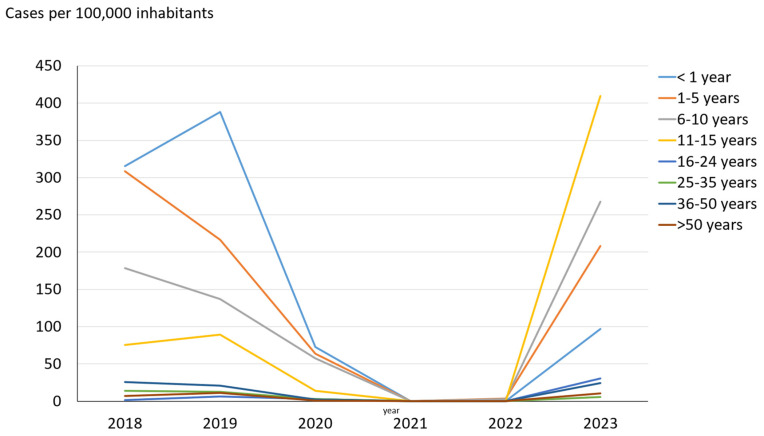
Age group distribution of the incidence of whooping cough in Gipuzkoa, north of Spain, 2018–2023.

**Figure 2 vaccines-12-01192-f002:**
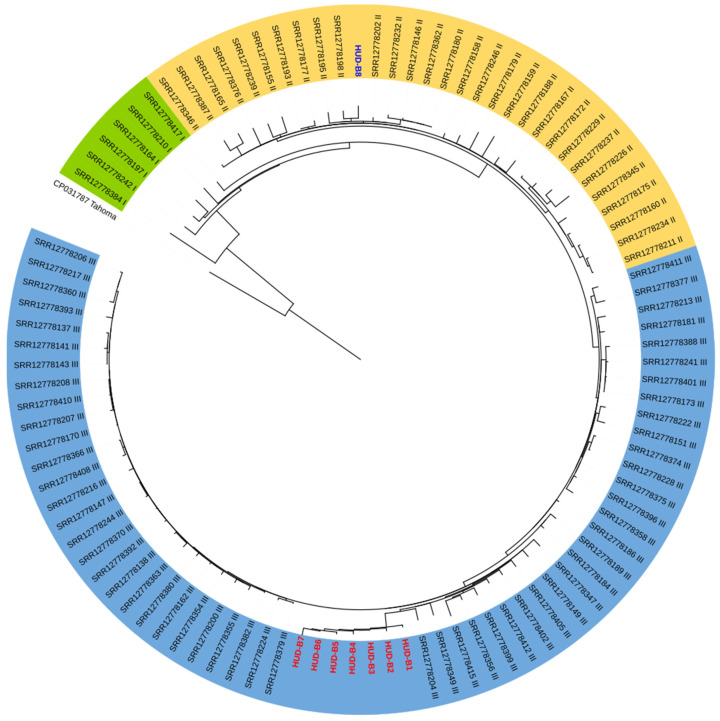
Maximum likelihood tree of *Bordetella pertussis* isolates from Spain (1998–2018), including 8 isolates from Gipuzkoa (in blue, HUD-B8 isolated in 1999; in red, HUD-B1 to HUD-B7 isolated in 2023).

**Table 1 vaccines-12-01192-t001:** Number of cases and incidence per 100,000 inhabitants of pertussis in Gipuzkoa, north of Spain, between 2018 and 2023.

Year	2018	2019	2020	2021	2022	2023	Total
Cases (no.)	241	219	51	0	2	371	884
Incidence	36.7	33.3	7.8	0.0	0.3	55.5	22.4

**Table 2 vaccines-12-01192-t002:** Pertussis vaccination status of the population of Gipuzkoa with confirmed *B. pertussis* infection in 2023.

Age in 2023 (in Years)	Correctly Vaccinated ^1,2^	Non-Vaccinated ^2^	Data Not Available	Total
<1	1 (100%)		3	4
1–4	20 (87.0%)	3 (13.0%)	1	24
5–10	96 (92.3%)	8 (7.7%)	2	106
11–15	137 (98.6%)	2 (1.4%)	11	150
16–24	19 (100%)			19
25–35			4	4
36–50		10 (100%)	23	33
>50		31 (100%)		31
Total	273	54	44	371

^1^ Four or five doses in individuals older than 6 years, 3 doses in children between 11 months and 6 years, 2 doses in children 4–6 months old, and 1 dose in children 2–4 months old. No vaccine was available for people older than 48 years. ^2^ Percentages calculated in patients in whom the vaccination status was available.

## Data Availability

Data are available upon request from the corresponding author. Genomic sequences were deposited in GenBank (accession numbers SAMN40551699 to SAMN40551706).

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
