# Peer review of "Pertussis Outbreak During 2023 in Gipuzkoa, North Spain"

_vaccines, 2024, doi:10.3390/vaccines12101192_

Round 1

Reviewer 1 Report

Comments and Suggestions for Authors

Apart from a few errors in the use of the English language and not using italics all the time when naming the bacterium, I do not see any major flaws in this manuscript. It would have been highly desirable to have a larger number of isolates genotyped. 

Minor comments:

Line 229, "the culture of this fastidious bacteria", please explain this scientifically. 

Discussion, is there data available in terms of effectiveness of the Infanrix and Vaxelis vaccines? If so, please include this. Do these manufacturers report the number of boosts required?  

The manuscript lacks of a Conclusions section

MALDI-TOF conditions must be described

Comments on the Quality of English Language

Minor errors exist throughout the document

Author Response

Note: all lines in this review are referred to the reviewed version (R1) of the manuscript.

Reviewer 1

Comments and Suggestions for Authors

Apart from a few errors in the use of the English language and not using italics all the time when naming the bacterium, I do not see any major flaws in this manuscript. It would have been highly desirable to have a larger number of isolates genotyped. 

Answer. The English language has been reviewed and all bacterial names have been written in italics as commented by the reviewer.

Minor comments:

  1. Line 229, "the culture of this fastidious bacteria", please explain this scientifically. 

Answer. In line 235, “… fastidious bacteria” has been replaced with “… this bacteria with particular nutritional requirements”.

  1. Discussion, is there data available in terms of effectiveness of the Infanrix and Vaxelis vaccines? If so, please include this. Do these manufacturers report the number of boosts required?

Answer. According to the review comment, the effectiveness of the vaccines has been included. In line 199 it has been added: “…with vaccines that have shown an efficacy of 86-88% (20,21) , “ and two new references (20 and 21) have been added and the rest renumbered accordingly.

Only Vaxelis vaccine report the need of one booster dose at least 6 months after the first cycle of vaccination with 2-3 injections. 

  1. The manuscript lacks of a Conclusions section

Answer. The conclusion was the last paragraph. In line 238 we have added a title to this section. “5. Conclusion

  1. MALDI-TOF conditions must be described

Answer. MALDI-TOF conditions and score has been added in line 90-93, as follows: “MALDI-TOF was performed by direct plate testing. After drying the colony, 2 µl of matrix dissolved in acetonitrile was added and allowed to dry again. Analysis was performed using MALDI biotyper sirius system (Bruker Daltonik GmbH, Bremen, Germany) and the MBT Compass MSP library v. 4.1. For species level identification, a cut-off score of ≥2.0 was used.”

Comments on the Quality of English Language: Minor errors exist throughout the document

The English language has been reviewed.

Reviewer 2 Report

Comments and Suggestions for Authors

Ongoing epidemics of pertussis occur in many countries after the COVID-19 pandemic. This study was aimed to describe the pertussis outbreak in north Spain during 2023. In addition to the disease epidemiology, result of whole genome sequencing for isolates during the outbreak was included. The study design was proper, the finding is important and has an added value for understanding of ongoing epidemics of pertussis in the world. I have only a few comments which should be addressed first.

1. It would be important to discuss/speculate a bit why the neighbouring region had low number of pertussis in line with vaccines used, vaccination coverage, or difference in laboratory methods used, etc?

2. It seems all samples tested by PCR were also tested by culture. Unfortunately no culture result was described. It would be better to add the result although the focus of this study was not for diagnostic comparison.

3. Authors stated that the commercial kit of PCR can detect and differentiate B. pertussis from B. parapertussis and B. holmesii. It would be good to provide some details e.g. the target genes of PCR?

4. line 39, "... can also be responsible on rare occasion." This sentence seems not accurate. During the COVID-19, increased detection of B. parapertussis was noticed in several countries, such as Germany, France and the USA. This should be added.  

Author Response

Reviewer 2

Note: all lines in this review are referred to the reviewed version (R1) of the manuscript.

Comments and Suggestions for Authors

Ongoing epidemics of pertussis occur in many countries after the COVID-19 pandemic. This study was aimed to describe the pertussis outbreak in north Spain during 2023. In addition to the disease epidemiology, result of whole genome sequencing for isolates during the outbreak was included. The study design was proper, the finding is important and has an added value for understanding of ongoing epidemics of pertussis in the world. I have only a few comments which should be addressed first.

  1. It would be important to discuss/speculate a bit why the neighbouring region had low number of pertussis in line with vaccines used, vaccination coverage, or difference in laboratory methods used, etc?

Answer. According to the reviewer comment, in line 232 the following sentence has been added: “This alert could also explain the difference with the lower incidences reported in nearby provinces that shared the same vaccination schedules.”

  1. It seems all samples tested by PCR were also tested by culture. Unfortunately no culture result was described. It would be better to add the result although the focus of this study was not for diagnostic comparison.

Answer. In line 151 we have added: “Bacterial culture was performed in 249 of the 371 pertussis episodes of 2023, and B. pertussis was isolated in 19 of them (7.6%).”

  1. Authors stated that the commercial kit of PCR can detect and differentiate B. pertussis from B. parapertussis and B. holmesii. It would be good to provide some details e.g. the target genes of PCR?

Answer. In line 85-86 we have added: “B. pertussis was detected in the samples of all patients using a commercial real-time PCR (Viasure, Certest BIOTEC, SL, Zaragoza, Spain) that detects and differentiates B. pertussisB. parapertussis and B. holmesii targeting the insertion sequence (IS) IS481 for B. pertussis and B. holmesii,  hIS1001 for B. holmesii and  a region of the pIS1001 for B. parapertussis.”

  1. line 39, "... can also be responsible on rare occasion." This sentence seems not accurate. During the COVID-19, increased detection of B. parapertussis was noticed in several countries, such as Germany, France and the USA. This should be added.  

Answer. In line 39 the sentence has been changed as follows: “although other Bordetella species, such as Bordetella parapertussis and Bordetella holmesii, can also be responsible of pertussis (1,2).”

A new reference (reference 2) has been added and the rest renumbered in the text and in the reference list accordingly.

Bouchez V, Toubiana J, Guillot S, et al. Transient reemergence of Bordetella parapertussis in France in 2022. J Med Microbiol. 2024;73. doi: 10.1099/jmm.0.001843. PMID: 38995835.